# CircRNA—Protein Interactions in Muscle Development and Diseases

**DOI:** 10.3390/ijms22063262

**Published:** 2021-03-23

**Authors:** Shuailong Zheng, Xujia Zhang, Emmanuel Odame, Xiaoli Xu, Yuan Chen, Jiangfeng Ye, Helin Zhou, Dinghui Dai, Bismark Kyei, Siyuan Zhan, Jiaxue Cao, Jiazhong Guo, Tao Zhong, Linjie Wang, Li Li, Hongping Zhang

**Affiliations:** Farm Animal Genetic Resources Exploration and Innovation Key Laboratory of Sichuan Province, College of Animal Science and Technology, Sichuan Agricultural University, Chengdu 611130, China; zhengshuailong@stu.sicau.edu.cn (S.Z.); zhangxujia@stu.sicau.edu.cn (X.Z.); emmanuelodame123@stu.sicau.edu.cn (E.O.); xiaoli_xu@stu.sicau.edu.cn (X.X.); chenyuan@stu.sicau.edu.cn (Y.C.); yejiangfeng@stu.sicau.edu.cn (J.Y.); 2018202017@stu.sicau.edu.cn (H.Z.); 71317@sicau.cn (D.D.); kyeibismark6@gmail.com (B.K.); siyuanzhan@sicau.edu.cn (S.Z.); jiaxuecao@sicau.edu.cn (J.C.); jiazhong.guo@sicau.edu.cn (J.G.); zhongtao@sicau.edu.cn (T.Z.); wanglinjie@sicau.edu.cn (L.W.)

**Keywords:** muscle, circRNA, RNA-binding protein, molecular interactions, function

## Abstract

Circular RNA (circRNA) is a kind of novel endogenous noncoding RNA formed through back-splicing of mRNA precursor. The biogenesis, degradation, nucleus–cytoplasm transport, location, and even translation of circRNA are controlled by RNA-binding proteins (RBPs). Therefore, circRNAs and the chaperoned RBPs play critical roles in biological functions that significantly contribute to normal animal development and disease. In this review, we systematically characterize the possible molecular mechanism of circRNA–protein interactions, summarize the latest research on circRNA–protein interactions in muscle development and myocardial disease, and discuss the future application of circRNA in treating muscle diseases. Finally, we provide several valid prediction methods and experimental verification approaches. Our review reveals the significance of circRNAs and their protein chaperones and provides a reference for further study in this field.

## 1. Introduction

Muscle is the primary animal tissue consisting of the directional differentiated muscle cells. According to muscle cells’ shape and distribution, muscles can be grouped into skeletal muscle, cardiac muscle, and smooth muscle [1]. Muscle is the power source for body and limb movements with contractile properties. Moreover, muscles also play critical roles in digestion, breathing, circulation, and excretion in mammals. Myogenesis is a complicated physiological process, including myoblast proliferation and differentiation alongside myotubes and muscle fibers’ formation. It lasts throughout the whole lifespan [2]. Myogenesis is precisely orchestrated by a variety of transcriptional factors such as the paired-homeobox family [3], myocyte enhancer factor 2 [4], and myogenic factor 5 [5], as well as muscle-related signaling pathways [6,7,8,9]. Moreover, with the deepening of research, some noncoding RNAs, including lncRNA (long noncoding RNA), miRNA (microRNA), and circRNA (circular RNA), are indispensable in the process of myogenesis [10,11,12]. 

Circular RNA (circRNA) is a new type of circular endogenous RNA formed by back-splicing of mRNA precursor [13]. In 1976, circRNA was firstly found in potato tuber viroid [14]. In the past two decades, limited by molecular biological technology, it is difficult for circRNA isolation and verification. On the basis of cognition, circRNA was once regarded as an abnormal splicing product formed in RNA transcription. It only exists in a small number of pathogens such as hepatitis virus and plant viroid, with no biological function [15,16]. With the development of high throughput sequencing technology and bioinformatics, it has been found that circRNAs widely exist. In 2012, Salzman et al. firstly identified 80 circRNAs in human normal nuclear malignant muscle disease cells through transcriptome sequencing and experimental identification, reporting that circRNA would be produced during gene transcription [17]. Ivano et al. launched RNA-Seq to detect RNA expressions of mouse and human myoblasts during the proliferation and differentiation period for the first time. The authors also witnessed the different circRNA expressions in the myoblast differentiation process compared with Duchenne muscular dystrophy and physiological state [18]. In recent years, many studies have found that circRNA can act as a miRNA molecular sponge or translate proteins to regulate muscle growth and development [11,18,19]. The latest research has found that circRNA can also regulate myogenesis by interacting with RBPs [20].

RNA-binding proteins (RBPs) are a type of protein dynamically interacting with RNAs through their specific spatial domain [21]. According to their unique domains, RBPs are classified into different types, such as zinc-finger (Znf), dsRNA-binding motif (dsRBM), K homology (KH), and RNA recognition motif (RRM) domains [22]. RBPs widely exist in animals, plants, and some microorganisms, involving biological processes such as cell proliferation, differentiation, migration, and apoptosis [21]. Previous studies have shown that RBPs can directly or indirectly regulate the function of circRNA by controlling the production and degradation of circRNA, helping circRNA transport between nucleus and cytoplasm, locating circRNA, and regulating circRNA translation [23,24]. 

In this review, we first discuss the molecular mechanism of circRNA–protein interaction and summarize the latest research on circRNA interacting with RBPs to regulate muscle development and related diseases. Finally, we sum up the bioinformatics prediction methods and experimental verification methods to provide support for revealing the mechanisms of circRNA–RBP interaction.

## 2. Characterization of CircRNAs and RPBs

### 2.1. CircRNAs

#### 2.1.1. Biogenesis of CircRNAs

CircRNAs are endogenous RNAs formed by particular back-splicing ways from mRNA precursors. According to their composition, they can be divided into circular intronic RNAs (ciRNAs) circulated of introns, exon circRNAs (EcircRNAs) consisting of exons, and exon–intron circRNAs (EIciRNAs) composed of introns and exons [25]. They exhibit distinct spatial distribution—ciRNAs and EIciRNAs mainly distribute in the nucleus, whereas EcircRNAs are primarily located in the cytoplasm, or even exosomes, through complex transport mechanisms (Figure 1A) [26,27,28]. Circulation of EIciRNA and EcircRNA are generally driven by intron pairing (Figure 1B), RBPs (Figure 1C), or lariat (Figure 1D) [13,29]. Among them, the occurrence of cyclization driven by intron pairing is more common than that of lariat structure [30]. When circulating by intron pairing, long flanking introns containing short repeat sequences, such as *Alu*, can promote the circularization of circRNA [13,28]. The formation of ciRNA mainly relies on the GU-rich motif (located in 5′ splice site, 7 nt) and the C-rich motif (found in branchpoint site, 11 nt). They form a lariat structure via polymerase II and finally cyclize through the covalent linkage of 2′,5′-phosphodiester bond (Figure 1E) [27].

#### 2.1.2. Properties of CircRNAs

In recent years, with the continuous development of high-throughput sequencing, many circRNAs have been identified in various tissues and cells. Due to its unique formation and spatial structure, circRNA has different biological characteristics from linear mRNA transcripts, such as (1) high stability—circRNAs are covalent closed circular RNAs without 5′-caps and 3′-poly(A) tails. As a result, compared with linear mRNA transcripts, circRNA has a solid tolerance to ribonuclease R (RNase R) [31]. (2) Pervasiveness—circRNA is widely present in cells and tissues [13]. The expression abundance of circRNA is generally low. However, some special circRNAs expressed much higher (≥10 times) than their linear transcript. For instance, circHIPK3 originated from the second exon of *HIPK3* gene, with higher levels than *HIPK3* mRNA [32]. (3) Conservation—circRNA is highly conserved among different species. It has been found that 15% to 20% of the splice sites of circRNAs in the mouse brain are highly conserved with those in pigs [33]. (4) Spatial, temporal, and tissue cell specificity—studies on *Drosophila*, mice, and humans unveil that circRNAs are highly enriched in the nervous system [34,35]. Moreover, RNA-Seq data of *pig* brain tissues suggest that circRNAs are dynamically expressed during embryonic development and most enriched at embryonic 60 days [33].

#### 2.1.3. Regulating Mechanisms of CircRNAs

CircRNAs exist widely in mammals and regulate genes at the transcriptional and post-transcriptional levels through various means. Current studies have found that circRNA mainly functions through the following mechanisms: (1) miRNA sponges (Figure 1F). CDR1as, as a classic circRNA molecular, can promote differentiation of goat skeletal muscle satellite cells through binding with miR-7 and internally competing with *IGF1R* mRNA [11]; circTTN promotes the proliferation and differentiation of bovine skeletal muscle satellite cells via competitively binding with mir-432, which functions to inhibit the expression of IGF2 [19]. (2) Regulating alternative splicing (Figure 1G). CircSEP3, generated from exon 6 of the *Sepallata3* (*SEP3*), combines with DNA locus to form an RNA–DNA hybrid (R-loop), and causes the suspension of *SEP3* transcription [36]. (3) Involving in gene transcription (Figure 1H). It has been widely known that some EIciRNAs can interact with U1 snRNA to form a complex EIciRNA-U1 small nuclear ribonucleoprotein (snRNP) to promote host gene expressions, such as circEIF3J and circPAIP2 [28]. However, the latest research has unveiled that ciRNAs possess the same function. For instance, ci-ankrd52 interacts with the elongation Pol II to promote the parent gene *ANKRD52* [27]. (4) Translation (Figure 1I). Many studies have reported that circRNA has coding potential [18,37,38]. (5) Pseudogene effect (Figure 1J). Dong et al. found that about 33 pseudogenes derived from circRFWD2 may regulate gene expression by changing the genome’s structure [39]. (6) Biomarker (Figure 1K). The abnormal circRNAs are connected with the occurrence and development of several diseases such as cancer. Thus, circRNAs could play a role as a biological molecular marker in diagnosing, treating, and prognosing of diseases [40]. (7) Interact with RBPs (Figure 1L). CircRNAs can specifically interact with protein molecules in cells and then control the expression of many downstream genes to regulate cell growth, differentiation, mobility, and apoptosis [20,41,42,43].

### 2.2. RBPs

RBPs are a class of protein molecules that can bind with single-stranded or double-stranded RNA. As essential participants in post-transcriptional regulation, they are involved in a series of biological processes, such as cell proliferation, differentiation, migration, and apoptosis, by regulating the expression of related genes [44,45].

#### 2.2.1. Number and Properties of RBPs

RNA-binding proteins are widely identified in animals, plants, and microorganisms, accounting for about 2%–8% of eukaryotic gene-coding proteins [21]. In the past, RBPs were predicted on the basis of the understanding of protein domains. By comparing protein sequences, researchers identified 500 RBPs in humans and mice [46,47,48]. Moreover, automatically functional annotation of Gene Ontology estimated that there are about 1900 RBPs in humans [49]. However, these traditional methods mentioned above are usually not accurate. False or omissive classification always occurs. With the development of high-throughput sequencing technology and molecular biology experimental technology, such as mass spectrometry, researchers have identified more than 1500 different RBPs in the human genome. These RBPs perform their corresponding biological functions by binding to other target RNA motifs [50]. Most RBPs generally have no tissue specificity. However, recent studies have found that certain RBPs are significantly tissue-specific. They are mainly enriched in the gonads, brain, muscle, bone marrow, and liver [50]. For example, RBPs Nova-1 and Nova-2, belonging to the Navo (neuro-oncological ventral antigen) family, were mainly expressed in the head and spinal cord. They contain three KH structure domains, which can combine with specific RNAs [51].

#### 2.2.2. RNA-Binding Domains of RBPs

RBPs have their specific RNA-binding domains (RBDs), recognizing and binding RNA motifs for post-transcriptional regulation [52,53]. Beyond that, these domains usually function as DNA binding and enzyme catalysis [54]. To date, identified RBDs mainly include RRM, zinc finger domain, KH domain, and dsRBM. RRM, zinc finger domain, and KH domain primarily bind with single-stranded RNA, whereas dsRBM could interact with double-stranded RNA [22,55,56,57]. In the process of evolution, RBPs need to continuously improve the specificity of sequence recognition to recognize more novel RNA motifs. As a consequence, one RBP often contains multiple RBDs [22].

## 3. RBPs Regulating CircRNA

Generally speaking, RBP–circRNA binding occurs through a specific or semi-conserved sequence of circRNA. However, unlike linear RNA, the distinct circularization mechanism of circRNA makes it reasonable to form more complex secondary and tertiary structures. For example, some circRNAs have loops, bulges, kinks, and other irregular structures. As a result, these specific spatial structures can be recognized explicitly by RBPs [42,43,58]. Mechanically, the interaction between RBPs and circRNAs is essential. The interaction between atoms, which can be divided into Van der Waals interaction, hydrogen bond interaction, electrostatic interaction, and so on, work together to stabilize the complex of RBPs and circRNAs [59].

Studies have suggested that RBPs also play essential roles in circRNA post-transcriptional regulation by dynamically forming a complex with circRNA, such as circRNA alternative splicing, modification, edition, transportation, degradation, translation, and other biological processes. Interestingly, the interaction of circRNAs and proteins may have bidirectional effects. CircRNAs can also serve as sponges, decoys, or scaffolds for proteins.

### 3.1. RBPs Regulate the Synthesis and Degradation of CircRNA

RBPs play essential roles in circRNA synthesis (Figure 2A). In *Drosophila* and human genomes, the *MBL/MBNL1* gene can form a circular RNA-circMBL, which depends on the directional binding of the RNA-binding protein MBL flanking intron-specific sequence [23]. Through genome-wide screening, Chen et al. found a series of trans-acting protein factors closely related to circular RNA generation and processing, including NF90 and NF110 related to antiviral immunity. The authors were able to promote the production of circRNAs by binding to the *Alu* motif on flanking intron via dsRBD [24]. In prostate cancer, heterogeneous nuclear ribonucleoprotein L (HRNPL) promotes circRNA formation by binding to exons’ flanking sequence [60]. Similarly, Quaking 1 (QKI) can also specifically bind to the flanking introns to encourage the formation of circRNA [61,62]. Fused in sarcoma (FUS) can also combine with exon or flanking intron to promote circRNA synthesis [63]. It has recently been reported that RNA-binding motif protein 20 (RBM20) is involved in the alternative splicing of the I-band region of the mouse *Titin* gene; after knocking out the RBM20 binding site, RBM20-related circRNAs cannot be generated [64]. Some RBPs, such as heterogeneous nuclear ribonucleoprotein (hnRNP) and serine-arginine (SR) protein, promote the production of circRNA [65], while some RBPs inhibit the formation of circRNA. For example, DEAH-box helicase 9 (DHX9) can bind to reverse complementary *Alu* elements, resulting in the uncoiling of *Alu* elements and inhibiting the formation of circRNA [66]. Moreover, adenosine deaminase 1 acting on RNA (ADAR1) specifically binds to and destabilizes *Alu* sequences. Consequently, it inhibits the generation of circRNA [67].

RBPs also play essential roles in circRNA degradation (Figure 2B). Due to its unique circular structure, circRNA is not affected by the classical linear mRNA degradation pathways like restriction exonuclease. However, there are few reports on the molecular mechanism of circRNA degradation. To date, the decay models of circRNA can be divided into four types. Interestingly, all of them require the participation of RBPs. (1) Degradation mediated by miRNA-AGO2. CDR1as are widely present in different species, which can specifically recruit mir-671, and then cause AGO2 binding protein to participate in the degradation of CDR1as [68]. (2) Degradation mediated by m^6^A modification. As is common knowledge, N^6^-methyladenosine (m^6^A) is the most common, abundant, and most conservative internal transcription modification, and it can also exist in all types of RNA, including circRNA. Some circRNAs modified by m^6^A can be recognized explicitly by YTHDF2 and HRSP12-binding proteins, which mediate endonuclease RNase P/MRP (mitochondrial RNA processing) and participate in the degradation of circRNA [69]. (3) Degradation mediated by PKR- RNase L. Some circRNAs also have RNA double-stranded structures, which can be specifically recognized by the protein kinase PKR-containing dsRBM domain. PKR mediates the degradation of circRNAs by RNase L [58]. (4) Degradation mediated by UPF1-G3BP1. Fischer et al. found that one-third of circRNA in the human species can form a highly integrated structure. UPF1 and G3BP1 can uniquely recognize these high-level structures and participate in the degradation of circRNA. The degradation pathway is dependent on the RNA-binding activity of UPF1 and G3BP1 to phosphorylate the S149 site, and it only occurs in circRNAs with higher structure [70].

### 3.2. RBPs Involved in the Modification and Editing of CircRNA

Recently, epigenetic alterations of circRNA, such as N^6^-methyladenosine (m^6^A) and N^5^-methylcytidine (m^5^C), have gained increased attention. As a unique RNA, epigenetic modifications of circRNA depend on a series of RBPs. For instance, m^6^A modification is widely found on circRNAs, such as circE7, circZNF609, depending on three different functional enzymes: “writer”, “eraser,” and “reader” (Figure 2C) [18,38,71]. These enzymes are composed of a series of RBPs, such as methyltransferase-like 3 and 14 proteins (METTL3 and METTL14), FAT and obesity-related proteins (FTO), and the YTH family of RNA-binding proteins [38]. Previous studies have reported that the m^6^A modification level of circRNA was affected after METTL3 knockdown [72]. Interestingly, m^6^A modification promotes the generation, degradation, and translation of circRNAs. The latest studies have also reported that m^5^C modification also exists in circRNA, but its molecular mechanism remains unknown [73].

### 3.3. RBPs Participate in the Transportation of CircRNA

CircRNA is produced in the nucleus, but it has a distinctive subcellular location, such as uniform distribution in the cytoplasm, ribosome, mitochondria, cytoplasm, and exosome. It relies on RBP-mediated transportation selective transportation mechanisms to play an essential role in various subcellular locations (Figure 2D). Studies have reported that the RNA-binding protein YTHDC1 will specifically recognize methylated circNSUN2, and YTHDC1 transfers circNSUN2 from nucleus to cytoplasm [74]. Moreover, similar phenomena appeared in *Drosophila* and humans. Huang et al. found that the interference of Hel25E in *Drosophila* significantly results in the enrichment of circRNA in the nucleus, determining that Hel25E in *Drosophila* mediates circRNA transportation; both UAP56 (DDX39B) and URH49 (DDX39A), the homologous proteins of Hel25E, can mediate circRNA nuclear export. UAP56 is mainly responsible for the nuclear exportation of sizeable molecular weight circRNA, whereas URH49 is primarily responsible for small molecule weight circRNA [75]. A large number of circRNAs containing 5′-GMWGVWGRAG-3ʹ motif in exosomes may be recognized and packaged into the exosomes by some RBPs recognizing this motif [76].

### 3.4. RBPs Control the Translation of CircRNA

It is generally believed that only mRNA can translate protein in eucaryon. However, the latest studies have demonstrated that some circRNAs have the coding ability. Although circRNAs lack m^7^G cap structure initiation mechanism like linear transcripts, it depends on a unique Cap-independent translation initiation mechanism, with the participation of RBPs during the initiation of translation (Figure 2E). Common Cap-independent translation initiation mechanisms mainly include internal ribosome entry site (IRES)-mediated translation initiation mechanism and m^6^A modification-mediated translation initiation mechanism. IRES-mediated translation of circRNA needs to rely on the IRES sequence in circRNA, a 150–250 bp sequence located upstream of the circRNA initiation codon. IRES can be folded into a transfer RNA (tRNA)-like structure and is recognized by the RNA-binding protein eIF4G2 (eukaryotic initiation factor 4G2) to drive circRNA translation [37,77]. For example, circPINT, circulated from lincPINT, could encode a PINT-87aa protein under IRES existence conditions to inhibit the occurrence of malignant glioma [78]. However, without IRES, some circRNA can also start translation after m^6^A modification. Yang et al. found that some conserved motifs existed in the UTR region of circRNAs. These motifs can be specifically recognized and modified by m^6^A methyltransferase and identified by YTHDF3 protein to form translation initiation complex eIF4 (eukaryotic initiation complex 4) [38]. Moreover, with m^6^A methylation, circE7 can translate E7 protein and regulate cancer cell proliferation [71]. 

## 4. CircRNAs Affecting RBPs

### 4.1. CircRNAs Can Serve as RBP Supermolecular Sponges

There are many binding motifs of miRNAs on circRNA, displaying a role of miRNA molecular sponge. Interestingly, many of these circRNAs contain one or more RBP binding sites, which can also act as molecular sponges to binding with proteins and function as endogenous competitive RNA (Figure 2F). For example, MBL protein can promote the synthesis of circMbl and inhibit the expression of its linear transcripts. However, excessive MBL can be sponged by circMbl to keep a dynamic balance of circMbl and its linear transcript [23]. Moreover, HuR combining of circPABPN1 results in decreased *PABPN1* mRNA stability and the reduction of PABPN1 protein expression [79].

### 4.2. CircRNAs Participate in the Transportation of RBPs

RBPs are diverted to different subcellular regions depending on complex transport pathways. CircRNAs play an irreplaceable role in this biological process (Figure 2G). For instance, circERBB2 has been proven to increase nucleolus localization of proliferation-associated 2G4 (PA2G4), promote interaction between PA2G4 and Pol I transcription factor (TIFIA), and recruit Pol I to their DNA promoter region [80]. Moreover, circSKA3 promotes TKS5 transfer to the cell membrane, forms a ternary complex with Integrin β1, and promotes invasive pseudopod formation [81]. Furthermore, mecciND1 and mecciND5 from mitochondria-encoded circRNAs (mecciRNAs) play essential roles in RBPs transport. MecciND1 interacts with nuclear-coding proteins replication protein A32 (RPA32) and replication protein A70 (RPA70) to facilitate their insertion into mitochondria. Meanwhile, MecciND5 promotes the entry of hnRNPA1, hnRNPA2B1, and hnRNPA2B1 into mitochondria [82].

### 4.3. CircRNAs Serve as Decoys for RBPs

Many circRNAs can also act as protein decoys of RBPs, promoting their transport between the nucleus and cytoplasm to maintain their specific functions (Figure 2H). For example, circAMOTL1 specifically transfers c-Myc proteins from the cytoplasm to the nucleus, promoting tumor formation [83]. Furthermore, circAMOTL1 can also transfer signal transducer and activator of transcription 3 (STAT3) from the cytoplasm to the nucleus to promote their combination *Dnmt3a* promoter and accelerate wound healing [84]. Moreover, circSTAG1 can act as a decoy for ALKBH5, specifically recruit ALKBH5, and inhibit its entry into the nucleus, thus changing the m^6^A modification of total RNA and improving the m^6^A modification level of total RNA [85].

### 4.4. CircRNAs Function as Scaffolds for RBPs

Many circRNAs in cells have multiple protein binding sites, which can bind various RBPs by acting as protein scaffolds to promote proteins contacting with each other (Figure 2I). For instance, circFoxo3 can regulate cell apoptosis sensitivity by competitively binding mouse double-minute 2 (MDM2) and p53. When circFOXO3 is mutated or treat by RNase R, it fails to function as a scaffold [86]. Meanwhile, circFOXO3 can also competitively bind with P21 and CDK2, promoting the inhibitory effect of p21 on CDK2 and participating in cell cycle regulation [41]. CircAMOTL1 can also act as a protein scaffold to competitively bind with 3-phosphoinositide-dependent protein kinase-1 (PDK1) and V-akt murine thymoma viral oncogene homolog (AKT), inhibiting cell apoptosis and promoting myocardial repair by promoting the phosphorylation and nucleation of AKT [87]. Moreover, circNDUFB2 could act as a scaffold to form a tripartite motif protein 25 (TRIM25)/circNDUFB2/IGF2BPs ternary complex, promoting ubiquitination degradation of IGF2BPs via enhancing the interaction between TRIM25 and IGF2BPs, as well as regulating cell growth [88].

## 5. CircRNA–Protein Interactions Regulating Muscle Development and Diseases

In recent years, more and more studies have found that noncoding RNA families, including circRNA, are widely involved in myoblast proliferation and differentiation, muscle formation, and the occurrence of muscle diseases. At present, muscle-related circRNA functions through complex mechanisms, including sponging miRNAs and encoding small peptides. For example, CDR1as competitively binds to miR-7 with IGF1R to promote skeletal muscle cell differentiation, whereas circZNF609 participates in regulating myogenic differentiation by translating small peptides [11,18]. The latest research has also found that circRNAs participate in the formation of muscles and the occurrence of muscle-related diseases by interacting with RBPs (Table 1).

### 5.1. Muscle Development

#### CircSamd4 and PURA/PURB

Myoblast differentiation is a complex physiological process regulated by a complex molecular network, including the PUR protein family. The PUR protein family comprises transcription factors purine-rich binding proteins alpha (PURA) and purine-rich binding proteins beta (PURB), which play an essential role in forming myotubules. They can bind to the promoter region of *MHC* to inhibit the transcription of *MHC* and MHC proteins’ production, thus inhibiting myogenic differentiation. CircSamd4 directly binds to PURA and PURB, blocking entry into the nucleus and indirectly promoting muscle cell differentiation (Figure 3A) [20].

### 5.2. Muscle Related Diseases

#### 5.2.1. CircAMOTL1 and ATK1/PDK1

Myocardial injury can induce senescence and death of cardiomyocytes and cardiac fibroblasts. Dead cells would disrupt typical ventricular structure and function via replacing fibrous scar. Zeng et al. found that circAMOTL1 is significantly enriched in neonatal cardiomyopathy tissue. Furthermore, circAMOTL1 induces V-akt murine thymoma viral oncogene homolog 1 (AKT1) phosphorylation and pAKT1 nuclear transport by combining AKT1 and PDK1. Activation of the AKT signaling pathway promotes cell survival and proliferation (Figure 3B). This study provides a theoretical basis for circAMOTL1 as a potential therapeutic agent for myocardial repair [87].

#### 5.2.2. CircFndc3b and FUS

Acute myocardial infarction (MI) has the characteristics of high morbidity and fatality rate, which will pose a severe threat to the life safety of patients, but its mechanism is still unclear. Garikipati et al. identified that circFndc3b is significantly downregulated in mice suffering from MI. In vivo and in vitro verification showed that circFndc3b negatively regulates myocardial cells and endotheliocyte apoptosis to improve myocardial function. Mechanically, circFndc3b functions not as a molecular sponge of miRNA but binds to the RNA binding protein FUS to positively regulate VEGF-A expression, ulteriorly improving the function reconstruction of myocardium after infarction (Figure 3C) [89].

#### 5.2.3. CircNFIX and Ybx1/Nedd4-l

After myocardial infarction and heart failure, cardiac function can be restored by promoting the proliferation of cardiomyocytes. By analyzing the circRNA expression profile data of neonatal and adult rat hearts, Huang et al. identified circNFIX, which was relatively conserved in humans, mice, and rats. Knockdown of circNFIX could promote proliferation and angiogenesis. Meanwhile, the apoptosis of cardiomyocytes was inhibited, thus improving myocardial function and prognosis. Further results confirm that circNFIX binds to Y-box binding protein 1 (Ybx1) to act as a protein scaffold and subsequently induce ubiquitination degradation of Ybx1 and inhibit cyclin A2 and cyclin B1 (Figure 3D) [90].

#### 5.2.4. CircYAP and TMP4/ACTG

Myocardial fibrosis is an important pathological feature of cardiac hypertrophy; therefore, inhibition of myocardial fibrosis is expected to be its clinical cure. The *YAP* gene produces circYAP, and it is expressed markedly low in fibrotic myocardial tissue. Through RNA pull-down test and mass spectrometry identification, circYAP can bind to RNA-binding proteins tropomyosin-4 (TMP4) and gamma-actin (ACTG). Subsequent studies unveiled that circYAP acts as a protein scaffold to promote TMP4 and ACTG complex formation to inhibit actin aggregation and thus inhibit fibrosis progression. After knockdown of circYAP, actin aggregation efficiency is enhanced and fibrosis accelerated, leading to cardiac hypertrophy’s pathological characteristics (Figure 3E) [91].

## 6. Strategies for Studying CircRNA–Protein Interactions

### 6.1. Bioinformatics Tools for CircRNA–Protein Interaction Prediction

Initially, researchers used the composition of amino acids and bases, RNA secondary structure, hydrogen bonds, Van der Waals tendency, and other features obtained from the analysis of the protein–RNA interaction interface to predict the interaction between circRNA and RNA-binding protein. In recent years, with the rapid development of bioinformatics and deep learning, researchers turn to use the three-dimensional structure information of RNA and protein and genome-scale information for prediction, which greatly improves prediction accuracy [92,93,94]. Among them, the prediction tools for predicting the interaction of circRNA and RNA-binding protein are mainly catRAPID [95], CRIP [96], RBPsuite [97], RBPmap [98], RPIseq [99], RPI-Pred [100], and CircInteractome [101].

Bellucci et al. used the combination of three physical and chemical characteristics of secondary structure tendency, hydrogen bond tendency, and Van der Waals tendency to predict protein–RNA interactions and developed catRAPID. Muppirala et al. developed an algorithm RPISeq on the basis of the composition of protein amino acids and RNA bases, which used the super vector machine (SVM) algorithm and the random forest algorithm to build a predictive model. Suresh et al. combined sequence and structure information to develop RPI-Pred to predict protein–RNA interactions. CRIP, based on convolutional neural network (CNN) and recurrent neural network (RNN), predicts RBP-binding sites on circRNAs using RNA sequences alone. RBPsuite divides the input circRNA sequence into 101-nucleotide fragments, uses CRIP to score the interaction between the elements and RBP, and further detects the binding fragments’ verified motifs, thereby giving the full-length sequence combine score distribution. Moreover, CircInteractome and RBPmap can also predict circRNA and its interacting proteins.

### 6.2. Experimental Approaches for CircRNA–Protein Interactions Verification

Nowadays, the most common research techniques for identifying circRNA–protein interaction are mainly based on immunoprecipitation alongside other analysis methods, including circRNA pull-down, RNA binding protein immunoprecipitation (RIP), RNase protection assay (RPA), fluorescence in situ hybridization (FISH), and tagged RNA affinity purification (TRAP).

#### 6.2.1. CircRNA Pull-Down and RIP

CircRNA pull-down is the most efficient way to detect the interaction between circRNA and its binding proteins. To date, there are two kinds of circRNA pull-down, namely, probe-recognized circRNA pull-down and in vitro cyclization circRNA pull-down. Probe-recognized circRNA pull-down depending on a biotin-labeled probe recognized explicitly to the target circRNA molecules in the Co-IP system [20]. Streptavidin-labeled magnetic beads can capture anything that interacts with the target gene, and the interacting proteins can be identified and analyzed by Western blot or mass spectrometry. The probe must cross the backspacing site and is recommended to be 30–40 nt in length. To eliminate the interference of linear transcripts, RNase R digestion is recommended. As for in vitro cyclization circRNA pull-down, it is generally similar to those mentioned above. The differences are that co-IP systems were launched depending on a biotin-labeled in vitro synthesized circRNA instead of a probe [102,103,104]. Due to the complex and unique structure of circRNA, rigorous controls must be designed in the experiment. Therefore, simultaneously overexpression and knockdown experiments are required to exclude non-specific signals effectively. 

Opposite with circRNA pull-down, RIP aims to detect whether the protein can bind with RNAs. To conduct RIP, an antibody is used for capturing RNA molecules in the Co-IP test, and subsequently combined with RNA-Seq or qPCR to identify enriched RNAs [105].

#### 6.2.2. RPA

RPA is a novel technique for studying RNA–RBP interactions [106]. If a DNA or RNA motif could combine with proteins, RNase H can hydrolyze the RNA molecule part that has no complementary pairing. According to this principle, we can design multiple DNA probes targeting circRNA. If circRNA interacts with the protein, the search will not bind accurately, and RNA will not be degraded by RNase H. This method can obtain the accurate binding domains of circRNA. In summary, RPA is a very effective way to study the RNA–protein interaction.

#### 6.2.3. FISH Co-Localization Assay

In situ hybridization is an essential technique for the study of nucleic acid subcellular localization. The co-localization analysis based on fluorescence in situ hybridization (FISH) and protein immunostaining (IHC) can visually reflect circRNA and protein localization. Therefore, we can preliminarily affirm the potential binding possibility of circRNA and target proteins [42].

#### 6.2.4. TRAP

TRAP is an effective method to detect the interaction between RNA and proteins. The RNA tagging system is mechanically introduced into a circRNA overexpression vector to express circRNAs with specific cell tags. CircRNA and its interacting target proteins are captured through precise and stable interactions between RNA tags and charged proteins. Then, cells are lysed and purified using glutathione affinity and magnetic agarose beads to obtain the captured protein–circRNA–target protein complex. Finally, the target proteins are isolated and purified. Further identification and analysis can be performed by mass spectrometry [43].

## 7. Perspectives

Compared with linear mRNA, circRNAs highly stabilize. They can avoid triggering TLR/RIG-I-mediated immune response without nucleotide modification, providing a theoretical basis for circRNAs to become disease markers, therapeutic targets, and therapeutic tools [107]. Recently, with the emergence of CRISPR-Cas13d screening tools, lipid nanoparticle (LNP) delivery system, and the realization of in vitro engineering preparation of circRNA, rapid application of circRNA in clinical therapy has become possible [108,109,110]. Currently, a few studies have preliminarily used circRNA for disease treatment. Lavenniah et al. constructed a circRNA sponge (circmiR) that can target the degradation of miR-132/212 family in vitro; after delivering circmiR to cardiomyocytes, the hypertrophy is weakened, and cardiac function is stable [111]. Moreover, Yang et al. used rabies virus glycoprotein-circSCMH1-extracellular vesicles to selectively deliver circSCMH1 to the brain; circSCMH1 binds to methyl-CpG binding protein 2 (MeCP2) and blocks MeCP2 transporting to the nucleus, promoting the expression of downstream target genes of MeCP2, enhancing neuronal plasticity, and effectively improving the functional recovery after stroke in mice and monkeys [112]. In muscle-related diseases, multiple circRNAs, including circFndc3b and circYAP, participate in regulating myocardial diseases by binding RBPs and promoting the recovery of cardiac function. In the future, circRNAs with high purity and specificity can be prepared in vitro and delivered to injured heart tissues with the LNPs delivery system’s help to improve heart function, which will open a new chapter for treating diseases in the heart.

## 8. Conclusions

Research on muscle growth and development has always been a hotspot of life sciences. With experimental techniques and our understanding, more and more muscle-related molecular networks have become investigated in recent years. It is a complex regulatory system that includes many genes, transcription factors, cytokines, and noncoding RNAs. Nevertheless, disturbance of the regulatory network can lead to the occurrence of muscle-related diseases. circRNAs play an essential role in this biological process. They can act as miRNA molecular sponges and translate proteins to regulate myogenesis. The latest research has shown that they can also function by binding RBPs. Interestingly, circRNA sometimes regulates myogenesis through multiple pathways. For example, circNFIX can bind to Ybx1 and Nedd4-1 and covalently competitive bind miR-214 to regulate cardiomyocyte proliferation.

The current research on the interaction between circRNA and RNA-binding protein is still in its infancy, and many problems remain unknown. For example, (1) currently, few biometric tools are available for circRNA and protein interaction prediction, and their accuracy is low. (2) These databases lack corresponding information on the interaction between circRNA and RNA-binding protein. (3) The experimental methods still need to be improved. For example, the circRNA pull-down practical technology is not mature enough. Off-target effects exist in biotin-labeled circRNA pull-down; meanwhile, the preparation of circRNA by cyclization in vitro is complex. (4) The nonconservation of proteins among animals causes a lack of a protein database for non-model animals. Hence, the result from mass spectrometry cannot be accurately annotated.

In the future, more and more circRNA and its RBPs will be identified. With the further understanding of circRNA and RBPs, the relationship between circRNA and its binding proteins will be gradually evident. Their molecular network in regulating muscle development and muscle-related diseases will become more and more perfected.

## Figures and Tables

**Figure 1 ijms-22-03262-f001:**
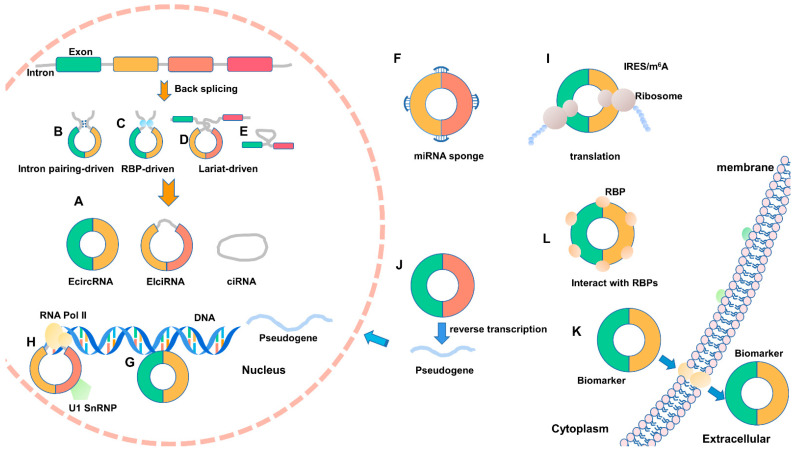
The biogenesis and regulating mechanisms of circular RNA (circRNA). (**A**–**E**) CircRNAs are formed by back-splicing into three major types of circRNA. (**F**–**I**) Various regulation mechanisms of circRNAs. (**J**) CircRNA can generate pseudogene by reverse transcription. (**K**) CircRNA can play a role as a biological molecular marker. (**L**) CircRNAs can interact with RBPs. IRES: internal ribosome entry site; m^6^A: N^6^-methyladenosine; U1 snRNP: U1 small nuclear ribonucleoprotein.

**Figure 2 ijms-22-03262-f002:**
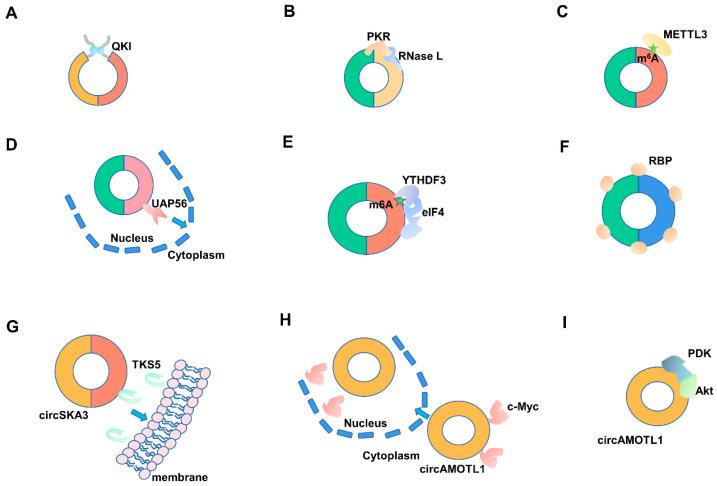
The interactions between circRNAs and RNA-binding proteins (RBPs). (**A**,**B**) RBPs regulate the synthesis and degradation of circRNA. (**C**) RBPs are involved in the modification and editing of circRNA. (**D**) RBPs participate in the transportation of circRNA. (**E**) RBPs control the translation of circRNA. (**F**) CircRNAs can serve as RBP supermolecular sponges. (**G**) CircRNAs participate in the transportation of RBPs. (**H**) CircRNAs serve as decoys for RBPs. (**I**) CircRNAs function as scaffolds for RBPs. m^6^A: N^6^-methyladenosine.

**Figure 3 ijms-22-03262-f003:**
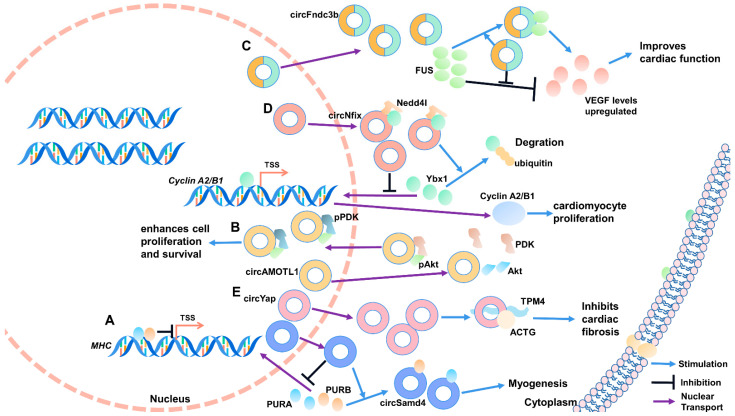
CircRNA–protein interactions regulating muscle development and diseases. (**A**) PURA and PURB inhibit the transcription of *MHC*, and circsamd4 promotes myogenesis by binding PURA and PURB. (**B**) circAMOTL1 induces AKT1 phosphorylation and pAKT1 nuclear transport by combining AKT1 and PDK1. (**C**) FUS negatively regulates VEGF-A expression, and circFndc3b promotes VEGF-A expression by binding to FUS. (**D**) circNFIX acts as a protein scaffold to enhance the binding of Ybx1 and Nedd41, induce ubiquitination degradation of Ybx1, and inhibit cyclin A2 and cyclin B1. (**E**) circYAP can inhibit cardiac fibrosis by acting as a protein scaffold to promote TMP4 and ACTG complexes’ formation. TSS: transcription start site.

**Table 1 ijms-22-03262-t001:** Summary of circRNAs and their protein chaperones in muscle development and diseases.

CircRNA	RBP	Function	Cell	References
CircSamd4	PURA/PURB	Myogenesis ↑	C2C12 myoblasts	[20]
CircAMOTL1	ATK1/PDK1	Cell proliferation and survival ↑	YPEN and MCF-7	[87]
CircFndc3b	FUS	Angiogenesis ↑ Cardiomyocyte apoptosis ↓	MCECs	[89]
CircNFIX	Ybx1/Nedd4-1	Cardiomyocyte proliferation ↓	Cardiomyocytes	[90]
CircYAP	TMP4/ACTG	Cardiac fibrosis ↓	MCFs and HL-1	[91]

Abbreviations: PURA, purine-rich binding proteins alpha; PURB, purine-rich binding proteins beta; AKT1, V-akt murine thymoma viral oncogene homolog 1; PDK, phosphoinositide-dependent kinase 1; FUS, RNA binding protein fused in sarcoma; Ybx1, Y-box binding protein 1; Nedd4-1, an E3 ubiquitin ligase; TMP4, tropomyosin-4; ACTG, gamma-actin; YPEN, rat endothelial cell line; MCF-7, mouse cardiac fibroblast; MCECs, mouse cardiac endothelial cells; MCFs, mouse cardiac fibroblasts; HL-1, mouse cardiomyocytes; ↑, increase/promotion; ↓, decrease/inhibition.

## Data Availability

No new data were created or analyzed in this study. Data sharing is not applicable to this article.

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
