# Peer review of "CircRNA—Protein Interactions in Muscle Development and Diseases"

_ijms, 2021, doi:10.3390/ijms22063262_

Round 1

Reviewer 1 Report

The manuscript by Zheng described recent findings of impacts of circRNA-RBP interactions in muscle development and diseases. This review brings together a wide body of information that may be of use for researchers interested in this field. My specific comment for this manuscript is as below.

As the title suggests, I believe the main theme of this review is the effect of cicRNA-RBP interactions on muscle differentiation and the development of muscle disease. Therefore, the authors should summarize the contents of Section 5 in some figures and/or tables. It will allow readers to understand at a glance the importance of the interaction between circRNA and RBP, and I believe that this review will be even more meaningful.

Author Response

We greatly appreciate your encouraging comment and thank you for your insightful suggestions.

Reviewer 2 Report

In this manuscript, Zheng and coauthors review the current knowledge on circRNA-RBP interactions in muscle physiology. The authors highlight the importance of RBPs in circRNA biogenesis and turnover. In addition, they summarized the role of circRNAs as RBP sponges in muscle development and disease by regulating RBP localization and function. Overall, the review is interesting and new to the field.

However, I have a few comments to improve the manuscript.

  1. Figure 2: This figure is more of general circRNA-RBP interaction. The figure should indicate the implications of the mentioned circRNA-RBP interaction in muscle physiology.
  2. The authors may include a table for the circRNAs, associated-RBPs, and their role in muscle physiology.
  3. Since the title talks about muscle development and disease, the article may include a section discussing the therapeutic implications of the described circRNA-RBP interactions in muscle diseases.
  4. For better understanding, some sentences need to be rewritten, especially in the abstract.

Author Response

Thank you for your comments concerning our manuscript; these comments are valuable and very helpful for revising and improving our paper. We have studied the comments carefully and have made revisions which we hope meet with approval.

Round 2

Reviewer 2 Report

The authors have revised the manuscript and improved it significantly. However, I have a minor comment to improve it further.

1. The "5.2.5. Perspectives" should be placed as a different section before the conclusion section as "7. Perspectives"

Author Response

We greatly appreciate your encouraging comment and thank you for your insightful suggestion.
